# Fermionic Greybody Factors in Schwarzschild Acoustic Black Holes

Sara Kanzi [1] and İzzet Sakallı [2],*

1  Faculty of Engineering, Final International University, North Cyprus via Mersin 10, Kyrenia 99320, Turkey
2  Physics Department, Eastern Mediterranean University, North Cyprus via Mersin 10, Famagusta 99628, Turkey
*  Correspondence: izzet.sakalli@emu.edu.tr

**Abstract:** In Schwarzschild acoustic black hole (SABH) spacetime, we investigate the wave dynamics for the fermions. To this end, we first take into account the Dirac equation in the SABH by employing a null tetrad in the Newman–Penrose (NP) formalism. Then, we consider the Dirac and Rarita–Schwinger equations, respectively. The field equations are reduced to sets of radial and angular equations. By using the analytical solution of the angular equation set, we decouple the radial wave equations and obtain the one-dimensional Schrödinger-like wave equations with their effective potentials. The obtained effective potentials are graphically depicted and analyzed. Finally, we investigate the fermionic greybody factors (GFs) radiated by the SABH spacetime. A thorough investigation is conducted into how the acoustic tuning parameter affects the GFs of the SABH spacetime. Both the semi-analytic WKB method and bounds for the GFs are used to produce the results, which are shown graphically and discussed.

**Keywords:** acoustic black holes; greybody factors; fermions; Dirac equation; Rarita-schwinger equation; Hawking radiation

## 1. Introduction

SABHs, also known as phonon black holes (BHs) or sonic BHs, are objects that are formed in certain types of fluids and exhibit some of the same characteristics as true BHs. These objects were first proposed by Unruh in 1981 [1,2], who demonstrated that the flow of a fluid through a converging nozzle could create an analogue of a BH horizon. Since then, there has been a great deal of research on SABHs, with a focus on understanding their properties and behavior. Today, we know that the theory obtained from the SABH metric has potential applications in various areas including: (i) the metric can be used to simulate acoustic BHs and study the behavior of sound waves in fluid systems, (ii) the concept of acoustic BHs can be used to understand the flow of fluids in pipes and other channels, (iii) the theory can be used to model BHs in space and study their properties and effects on surrounding objects, (iv) the study of acoustic BHs can help in understanding the principles of quantum mechanics in a classical context, and (v) the theory can provide insights into the thermal properties of BHs and the behavior of gases in confined spaces. Overall, the theory from the SABH metric has the potential to provide a deeper understanding of various physical systems and help in the development of new technologies in fields such as engineering and physics.

One of the areas of interest in the literature has been the investigation of the thermodynamic properties of SABHs [3]. It has been shown that these objects exhibit a Hawking-like temperature and entropy, suggesting that they may be subject to similar thermodynamic laws as ordinary BHs. Other researchers have focused on understanding the behavior of waves in the vicinity of SABHs [4,5], including the scattering and absorption of both phonons and photons, and gravitational lensing phenomena. There have also been a number of studies on the formation and stability of acoustic BHs [6,7], including the

role of dissipation and the influence of external perturbations. In addition, there has been a significant amount of work on the applications of acoustic BHs, including the simulation [8,9] of BH physics in the laboratory and the study of condensed matter systems [10,11]. Overall, the research on acoustic BHs will continue to contribute to our understanding of the behavior of matter and energy in the presence of BH-like objects and has potential applications in a range of fields [12].

The particle emission by BHs was studied in several articles (see Ref. [13] and references therein). Among them, there exist interesting ones such as Ref. [14], which shows that potential barriers can block the Hawking radiation (HR) [15] to some extent and even stop the radiation. In contrast, Koga and Maeda [16] showed by numerical computation that HR of the dilaton BH, for example, wins over the barrier and does not stop radiating, despite the fact that the potential barrier becomes infinitely high. In general, the thermal radiations of BHs have been conducted for the emission of bosons and in the semi-classical approach. As expected, fermionic (both spin- $\frac{1}{2}$ and - $\frac{3}{2}$) emission might show more or less similar properties qualitatively as scalars, but considering absorption or emission coefficients of the thermal waves that depend on the frequency of the radiation, GFs can change the scene and might become the key factor when characterizing and detecting BHs. Meanwhile, in physics, a greybody is an object or system that emits or absorbs electromagnetic radiation (EM (electromagnetic) radiation) in a manner that is dependent on the frequency of the radiation but independent of the direction of the radiation. The term "grey" comes from the fact that such an object or system is not a perfect absorber or emitter of EM radiation. For example, a perfect absorber (a blackbody) would absorb all of the incident EM radiation, regardless of its frequency. On the other hand, a perfect emitter, the so-called white body, reflects all incident radiation, as opposed to the black body that absorbs it all. A greybody, however, has an absorption or emission coefficient that depends on the frequency of the radiation. In cosmology, GF is a correction factor that appears in the calculation of the temperature and flux of radiation emitted or absorbed by a body in outer space, such as a BH, a neutron star or a dark matter candidate. It can be represented as the ratio of actual radiated power to the radiated power by a blackbody of the same temperature, as a function of frequency. The GF is a complex number, with a magnitude less than or equal to one and a phase that is dependent on the particular system being considered.

It was quickly realized that following the discovery of HR [15], BHs that are large and have a large ratio of electric charge to mass ($\frac{e}{m} \simeq 2 \times 10^{21}$) are unlikely to retain their charge and will quickly lose it through radiation. As a result, neutral BHs are more likely to occur in nature, rather than charged ones. This was pointed out in the work by Gibbons [17]. Both numerical and analytical methods have been used to calculate the rate of radiation emitted by fermions by using the GFs in the semi-classical approximation for particle with spin-1/2 [18–27] and particle with spin-3/2 [28,29]. Many studies have also discussed the scattering parameters of the fermionic field, including quasinormal frequencies, in the context of various BH spacetimes [30–48] and also for acoustic BH [49–51].

The goal of this study is to address the lack of fermionic GF solutions of Dirac and Rarita–Schwinger fields in the spacetime of SABH for all possible modes. An effort will be made to improve the accuracy of our analysis by employing semi-analytic bounds. Appropriately chosen ansatzes for the wave functions will allow us to derive the radial equations of the Dirac and Rarita–Schwinger fields in the background of SABH. Analytical expressions for the effective potentials will be obtained and, in the sequel, the asymptotic low-energy values of the GF will be found in each case. The role of the tuning (acoustic) parameter on the GFs will also be examined. The outline of our paper is as follows: in Section 2, we present the SABH under consideration and observe its some physical features. In Section 3, we focus on the propagation of spin- $\frac{1}{2}$ fields on the SABH: we derive the radial equation of the massless Dirac equation for arbitrary modes and obtain the corresponding effective potentials. In Section 4, the whole procedure is repeated for a Rarita–Schwinger

field propagating in the SABH geometry. Section 5 is devoted to the GF analysis of the fermions in the SABH geometry. We draw our conclusions in Section 6.

## 2. SABH Spacetime

As stated above, the concept of a SABH metric in fluid dynamics was first introduced by Unruh in 1981 [1]. Unruh's work was based on the observation that the equations of fluid dynamics can be transformed into the form of the equations governing the behavior of fields in a curved spacetime. He proposed that the analog of a BH in fluid mechanics can be created by a flowing fluid with supersonic velocity. Namely, Unruh showed that under certain conditions, sound waves in a fluid can be described by the same equations that describe the behavior of fields in a curved spacetime near a BH. This led to the development of the concept of an "acoustic BH", where the fluid flow plays the role of the gravitational field, and the speed of sound in the fluid plays the role of the speed of light. Since the introduction of the acoustic BH metric, it has been the subject of extensive research and has been applied in a variety of areas, including acoustics, fluid dynamics, astronomy, quantum mechanics, and thermodynamics.

In this section, we shall introduce the acoustic BH in a four-dimensional Schwarzschild framework, which can be considered as one of the simplest analogue BHs in curved spacetime. The action for the SABH solution is given in the relativistic Gross–Pitaevskii theory [52] as follows

$$S = \int d^4 x \sqrt{-g} (|\partial_\mu \varphi|^2 + m^2 |\varphi|^2 - \frac{b}{2} |\varphi|^4), \tag{1}$$

where $b$ denotes a constant parameter, and $m^2$ represents a temperature-dependent parameter as $m^2 \sim (T - T_c)$. $\varphi$ is a complex scalar field, which satisfies the following equation:

$$\Box \varphi + m^2 \varphi - b |\varphi|^2 \varphi = 0. \tag{2}$$

By considering the background spacetime as the Schwarzschild BH metric:

$$\begin{aligned} ds_{bg}^2 &= g_{tt} dt^2 + g_{rr} dr^2 + g_{\vartheta\vartheta} d\vartheta^2 + g_{\phi\phi} d\phi^2 \\ &= -f(r) dt^2 + \frac{dr^2}{f(r)} + r^2 (d\vartheta^2 + \sin^2\vartheta d\phi^2), \end{aligned} \tag{3}$$

with the metric function $f(r) = 1 - \frac{2M}{r}$ in which $M$ is the mass of the BH and making some straightforward computations, Guo et al [3] derived the SABH line-element as follows

$$ds^2 = \sqrt{3} c_s^2 \left[ -F(r) dt^2 + \frac{dr^2}{F(r)} + r^2 (d\vartheta^2 + \sin^2\vartheta d\phi^2) \right], \tag{4}$$

in which $c_s^2$ denotes the sound velocity, which can be set as $c_s^2 = 1/\sqrt{3}$ without loss of generality [4]. The metric function $F(r)$ of SABH is given by

$$F(r) = \left(1 - \frac{2M}{r}\right) \left[1 - \xi \frac{2M}{r}\left(1 - \frac{2M}{r}\right)\right], \tag{5}$$

where $\xi$ is a positive tuning parameter. One can immediately observe that metric (4) reduces to the Schwarzschild BH (3) as $\xi \to 0$.

There are three different solutions for $F(r) = 0$: $r_{bh} = 2M$ and $r_{ac_\pm} = (\xi \pm \sqrt{\xi^2 - 4\xi})M$, in which $r_{bh}$ represents the optical and $r_{ac_\pm}$ represent the outer $(+)$ and inner $(-)$ acoustic event horizons. To make the analysis in the existence of the acoustic event horizons region, let us consider $\xi \geq 4$. For $\xi = 4$, the acoustic BH becomes extremal with the clashed horizons of $r_{ac_-} = r_{ac_+} = 4M$. Moreover, if $\xi \to \infty$, then $r_{ac_+} \to \infty$, which means that there is no way for the sound to leave the spacetime.

In the case of $\xi \geq 4$, the spacetime has four regimes: (1) $r < r_{bh}$ represents the inside of the BH; (2) $r_{bh} < r < r_{ac_-}$ and (3) $r_{ac_-} < r < r_{ac_+}$, which both (2) and (3) regimes mean

that the sound cannot escape the BH but light can; and (4) in the regime of $r > r_{ac_+}$, both light and sound could escape from the BH. On the other hand, we will consider the outer horizon when obtaining the thermodynamic properties of SABH.

The Hawking temperature of an acoustic BH is a fundamental concept in the study of fluid dynamics and acoustics. It refers to the thermal radiation that is emitted by an acoustic BH, similar to the radiation emitted by a real BH. The concept of Hawking temperature was first introduced by S. Hawking in 1974 [53], who showed that BHs emit thermal radiation as a result of quantum mechanical effects near the event horizon. This radiation is known as HR. In the case of acoustic BHs, the analog of the event horizon is a sonic horizon, which separates the regions of subsonic and supersonic flow in a fluid. The temperature of the emitted thermal radiation can be calculated by using the concept of the Hawking temperature, which is proportional to the surface gravity at the sonic horizon. Recent studies in the literature [7,54–56] have shown that the Hawking temperature of an acoustic BH depends on the properties of the fluid, such as its density, pressure, and speed of sound. The temperature also depends on the properties of the fluid flow, such as the velocity and acceleration of the fluid. Moreover, the concept of Hawking temperature has been applied in various areas, including the study of Bose–Einstein [54,55] condensates, the behavior of waves in fluid systems, and the thermal properties of BHs in astrophysics. It is also worth noting that a team of researchers led by J. Steinhauer reported observing quantum HR for the first time [57]. The team used a Bose–Einstein condensate, a type of superfluid, to simulate a BH and observed the emission of sound waves that correspond to the HR [58–60]. Those studies open up a new path for further studies of quantum HR and the understanding of BHs in the quantum realm [61,62]. To sum up, the study of the Hawking temperature of acoustic BHs has been a valuable tool in the understanding of the behavior of fluids and waves in a curved spacetime and has led to advances in the fields of acoustics, fluid dynamics, and physics.

At this stage, it is worth noting that the Hawking temperature of SABH can be obtained by using the statistical definition of the Hawking temperature [63], which is based on the definition of surface gravity $\kappa$ [15]:

$$T_H = \frac{\kappa}{2\pi} = \frac{1}{4\pi} \lim_{r \to r_+} \frac{\partial_r g_{tt}}{\sqrt{g_{tt} g_{rr}}}, \tag{6}$$

so that we have

$$T_{H+} = \frac{\xi \left( \xi^{3/2} \sqrt{\xi - 4} + \xi^2 - 3\sqrt{\xi}\sqrt{\xi - 4} - 5\xi + 4 \right)}{M\pi (\xi + \sqrt{\xi}\sqrt{\xi - 4})^4}. \tag{7}$$

The entropy of the SABH can be computed by using the first law of thermodynamics: $dM = \pm T_{H+} dS_{BH}$, in which $S_{BH}$ denotes the Bekenstein–Hawking entropy [64]:

$$S_{BH} = \frac{A}{4}, \tag{8}$$

where $A = 4\pi r_h^2$ is nothing but the surface area of the SABH. Hence, the explicit form of the entropy of the SABH reads

$$S_{BH} = \pi M^2 (\xi + \sqrt{\xi^2 - 4\xi})^2. \tag{9}$$

At this point, we would like to point out once again that the case of $\xi \geq 4$, which we have considered in this article, guarantees that the entropy (9) remains real positive. Figure 1 exhibits the behaviors of the Hawking temperature and Bekenstein–Hawking entropy with respect to the $\xi$ parameter. In particular, the temperature graph is in accordance with the Maxwell–Boltzmann distribution [65].

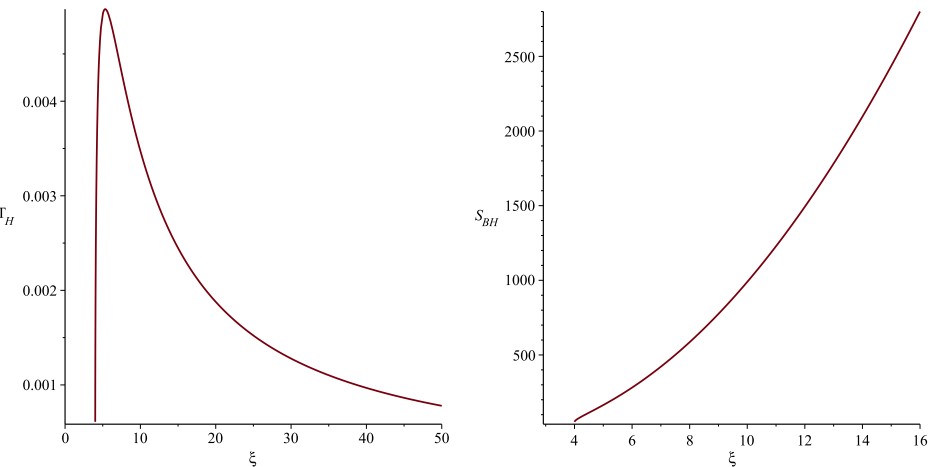

**Figure 1.** Graphs of Hawking temperature vs. $\xi$ (**left**) and Bekenstein–Hawking entropy vs. $\xi$ (**right**) for the SABH spacetime.

### 3. Dirac Equation

The Dirac equation [66] is a fundamental equation in quantum mechanics that describes the behavior of spin-$\frac{1}{2}$ particles, such as electrons. It is written as a first-order partial differential equation that describes the relationship between the wave function of a particle and its energy and momentum. In four-dimensional curved spacetime, the Dirac equation takes into account the effects of gravity on the behavior of particles. This is achieved by replacing the flat Minkowski spacetime metric used in the original form of the equation with a curved spacetime metric that describes the geometry of the spacetime in the presence of gravity. The Dirac equation in curved spacetime has been the subject of extensive research and has been used to study various phenomena in physics and astronomy, including the behavior of electrons in strong gravitational fields and the generation of gravitational waves [67]. One of the key results of this research has been the discovery that the Dirac equation in curved spacetime predicts the existence of fermions, a type of particle that makes up matter, and their antiparticles, known as antifermions. The equation also predicts the behavior of these particles in strong gravitational fields, which is important for understanding the behavior of matter near BHs and other astronomical objects. In addition to its applications in physics and astronomy, the Dirac equation in curved spacetime has also been used in the development of theories of quantum field theory and quantum gravity, where it is used to describe the behavior of particles and fields in the presence of gravity [68].

Our concentration in this section is to derive the effective potential of fermions with spin-$\frac{1}{2}$ propagating in the SABH geometry. To this end, a one-dimensional Schrödinger-type wave equation is aimed to be obtained by employing the massless Dirac equation having the Dirac field $\Psi$ [69].

$$\gamma^\alpha e_\alpha^\mu (\partial_\mu + \Gamma_\mu)\Psi = 0, \tag{10}$$

where $\gamma^\alpha$ and $\Gamma_\mu = \frac{1}{8}[\gamma^\alpha, \gamma^\beta]e_\alpha^\nu e_{b\nu;\mu}$ represent the Dirac matrix and spin connection, respectively, and $e_\alpha^\mu$ indicates the inverse of the tetrad $e_\mu^\alpha$ which is defined as

$$e_\mu^\alpha = diag(\sqrt{F}, \frac{1}{\sqrt{F}}, r, rsin\theta). \tag{11}$$

Therefore, Equation (10) can be rewritten as

$$-\frac{\gamma_0}{\sqrt{F}}\frac{\partial\Psi}{\partial t} + \sqrt{F}\gamma_1\left(\frac{\partial}{\partial r} + \frac{1}{r} + \frac{1}{4F}\frac{dF}{dr}\right)\Psi + \frac{\gamma_2}{r}\left(\frac{\partial}{\partial\theta} + \frac{1}{2}cot\theta\right)\Psi + \frac{\gamma_3}{rsin\theta}\frac{\partial\Psi}{\partial\varphi} = 0. \tag{12}$$

By considering the Dirac field as

$$\Psi = F^{-1/4}\Phi,\tag{13}$$

Equation (12) can be simplified to

$$-\frac{\gamma_0}{\sqrt{F}}\frac{\partial\Phi}{\partial t} + \sqrt{F}\gamma_1\left(\frac{\partial}{\partial r} + \frac{1}{r}\right)\Phi + \frac{\gamma_2}{r}\left(\frac{\partial}{\partial\theta} + \frac{1}{2}cot\theta\right)\Phi + \frac{\gamma_3}{rsin\theta}\frac{\partial\Phi}{\partial\varphi} = 0.\tag{14}$$

Applying the tortoise coordinate $dr_* = \frac{dr}{F}$ transformation and the following ansatz to Equation (14), one obtains

$$\Phi = \begin{pmatrix} \frac{iG^{(\pm)}(r)}{r}\phi_{jm}^{\pm}(\theta,\varphi) \\ \frac{H^{(\pm)}(r)}{r}\phi_{jm}^{\mp}(\theta,\varphi) \end{pmatrix}e^{-i\omega t},\tag{15}$$

in which

$$\Phi_{jm}^{+} = \begin{pmatrix} \sqrt{\frac{j+m}{2j}}Y_l^{m-1/2} \\ \sqrt{\frac{j-m}{2j}}Y_l^{m+1/2} \end{pmatrix}, \qquad (j = l + \frac{1}{2}),\tag{16}$$

and

$$\Phi_{jm}^{-} = \begin{pmatrix} \sqrt{\frac{j+1-m}{2j+2}}Y_l^{m-1/2} \\ -\sqrt{\frac{j+1+m}{2j+2}}Y_l^{m+1/2} \end{pmatrix}. \qquad (j = l - \frac{1}{2})\tag{17}$$

After decoupling the equations, one can obtain

$$\frac{d^2H}{dr_*^2} + (\omega^2 - V_1)H = 0,\tag{18}$$

$$\frac{d^2G}{dr_*^2} + (\omega^2 - V_2)G = 0,\tag{19}$$

where

$$V_1 = \frac{\sqrt{F}|k|}{r^2}\left(|k|\sqrt{F} + \frac{r}{2}\frac{df}{dr} - f\right), \qquad (k = j + \frac{1}{2}, j = l + \frac{1}{2}),\tag{20}$$

$$V_2 = \frac{\sqrt{F}|k|}{r^2}\left(|k|\sqrt{F} - \frac{r}{2}\frac{df}{dr} + f\right), \qquad (k = -(j + \frac{1}{2}), j = l - \frac{1}{2}).\tag{21}$$

Thus, the effective potentials of the fermionic waves having spin-$\frac{1}{2}$ and moving in the SABH geometry are found as

$$V_{eff} = \frac{k^2A}{r^2}\left(1 \pm \frac{1}{\sqrt{A}}\left(\frac{df(r)}{dr}(\frac{r}{2} - 2M\xi f(r)) + f(r)(\frac{3M\xi}{r}f(r) - 1)\right)\right),\tag{22}$$

where $A = \sqrt{f(r) - \frac{2M\xi}{r}f^2(r)}$ and positive and negative signs are conjugated with spin signs. The behaviors of the effective potentials are depicted in Figure 2 for various $\xi$ parameters.

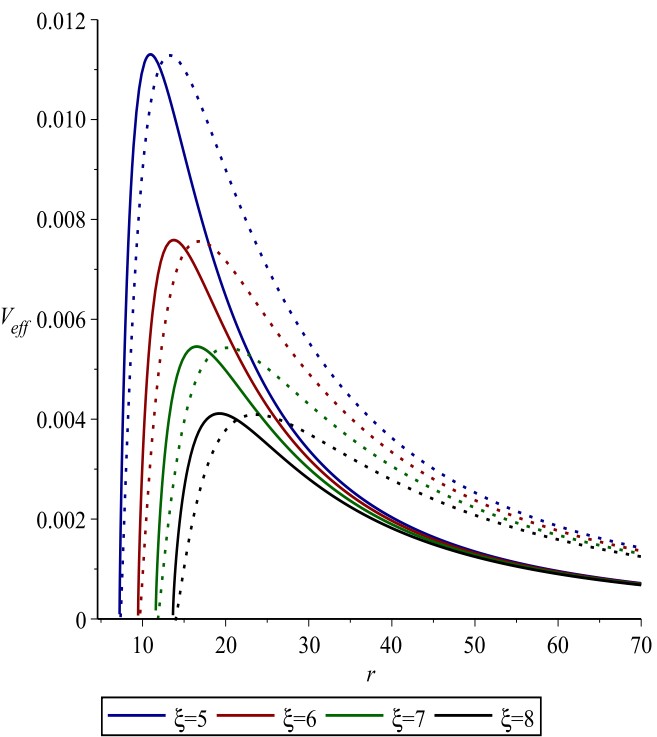

**Figure 2.** Plots of $V_{eff}$ versus $r$ for the spin-1/2 particles. The physical parameters are chosen as $M = j = k = 1$.

## 4. Rarita–Schwinger Equation

The Rarita–Schwinger equation was first derived by W. Rarita and J. Schwinger in 1941 [70], and it has since been an important tool for the study of high-spin particles and their interactions with other fields. In particular, it is used in the study of supersymmetric theories, where spin-$\frac{3}{2}$ fields appear as superpartners of spin-$\frac{1}{2}$ fields. The Rarita–Schwinger equation is a partial differential equation that describes the behavior of spin-$\frac{3}{2}$ fields in a four-dimensional curved spacetime; spin-$\frac{3}{2}$ fields, also known as Rarita–Schwinger fields, are fields that are characterized by having spin-$\frac{3}{2}$ [71,72]. Furthermore, in particle physics [73], quarks are elementary particles that have spin-$\frac{1}{2}$. However, composite particles made up of quarks can have different spins, including spin-$\frac{3}{2}$. Moreover, it is worth noting that mesons are formed by two quarks, a quark–antiquark pair. In the case of a meson made up of a quark and an antiquark with spins of $\frac{1}{2}$ and $\frac{-1}{2}$, respectively, their spins can combine in two possible ways: One possibility is that their spins cancel out, resulting in a composite particle with spin-0. This is the case for the pion $\pi^+$, which is made up of an up quark and a down antiquark. Another possibility is that their spins add up to create a composite particle with a higher spin value. In this case, the spins can combine in two different ways to produce a composite particle with spin-1 (vector bosons, which are governed by the Proca equation [74]). In a curved spacetime, the Rarita–Schwinger equation is modified to take into account the effects of gravity on the spin-$\frac{3}{2}$ fields. The equation becomes a covariant equation, meaning that its form is unchanged under general coordinate transformations. This is important for the consistency of physical predictions, as it ensures that the equation describes the behavior of the fields in a way that is independent of the choice of coordinates. The Rarita–Schwinger equation in a four-dimensional curved spacetime has been the subject of much research and has been applied in various areas, including particle physics, string theory, and cosmology. The equation is also related to other theories in physics, such as general relativity and Yang–Mills theory [75].

In this section, for the SABH spacetime, we shall consider the Rarita–Schwinger equation [76]:

$$\gamma^{\mu\nu\alpha}\tilde{D}_\nu\psi_\alpha = 0, \tag{23}$$

where $\psi_\alpha$ indicates the spin-3/2 field, and $\gamma^{\mu\nu\alpha}$ mentions the antisymmetric of Dirac matrices as

$$\gamma^{\mu\nu\alpha} = \gamma^\mu\gamma^\nu\gamma^\alpha - \gamma^\mu g^{\nu\alpha} + \gamma^\nu g^{\mu\alpha} - \gamma^\alpha g^{\mu\nu}. \tag{24}$$

In Equation (23), $\tilde{D}$ is the super-covariant derivative, which is defined for four-dimensional spacetime as

$$\tilde{D}_\mu = \nabla_\mu + \frac{1}{4}\gamma_\rho F_\mu^\rho + \frac{i}{8}\gamma_{\mu\rho\sigma}F^{\rho\sigma}. \tag{25}$$

At this stage, our concentration will be on the non-$TT$ eigenfunctions [29]; therefore, the radial and temporal wave functions are given as

$$\psi_r = \phi_r \otimes \bar{\psi}_{(\lambda)}, \qquad\qquad \psi_t = \phi_t \otimes \bar{\psi}_{(\lambda)}, \tag{26}$$

in which $\bar{\psi}_{(\lambda)}$ represents an eigenspinor with an eigenvalue of $i\bar{\lambda}$ where $\bar{\lambda} = j + 1/2$ and $j = 3/2, 5/2, 7/2, \ldots$. Moreover, the angular wave function is determined by

$$\psi_{\theta_i} = \phi_\theta^{(1)} \otimes \bar{\nabla}_{\theta_i}\bar{\psi}_{(\lambda)} + \phi_\theta^{(2)} \otimes \bar{\gamma}_{\theta_i}\bar{\psi}_{(\lambda)}, \tag{27}$$

where $\phi_\theta^{(1)}$ and $\phi_\theta^{(2)}$ depend on $r$ and $t$. It is worth mentioning that the specific selection of gamma tensors and spin connections used in this work can be found in Ref. [29]. By utilizing the Weyl gauge ($\phi_t = 0$) and using the same arguments used in [29], one can obtain the following gauge invariant variable

$$\Phi = -\left(\frac{\sqrt{F}}{2}i\sigma^3\right)\phi_\theta^{(1)} + \phi_\theta^{(2)}, \tag{28}$$

which can be rewritten as

$$\Phi = \begin{pmatrix} \phi_1 e^{-i\omega t} \\ \phi_2 e^{-i\omega t} \end{pmatrix}. \tag{29}$$

In Equation (29), parameters $\phi_1$ and $\phi_2$ are radially dependent, which can be defined as

$$\phi_1 = \frac{F - \bar{\lambda}^2}{B_1 F^{1/4}}\tilde{\phi}_1, \qquad\qquad \phi_2 = \frac{F - \bar{\lambda}^2}{B_2 F^{1/4}}\tilde{\phi}_2. \tag{30}$$

where $B_1 = \sqrt{F} - \bar{\lambda}$ and $B_2 = \sqrt{F} + \bar{\lambda}$. Now, with aid of the tortoise coordinate, we obtain a set of one-dimensional Schrödinger-like wave equations

$$-\frac{d^2}{dr_*^2}\tilde{\phi}_1 + V_1\tilde{\phi}_1 = \omega^2\tilde{\phi}_1, \tag{31}$$

$$-\frac{d^2}{dr_*^2}\tilde{\phi}_2 + V_2\tilde{\phi}_2 = \omega^2\tilde{\phi}_2, \tag{32}$$

whose potentials are given by

$$V_{1,2} = \pm F(r)\frac{dW}{dr} + W^2, \tag{33}$$

where

$$W = \frac{\bar{\lambda}\sqrt{F}}{r}\left(\frac{\bar{\lambda}^2 - 1}{\bar{\lambda}^2 - F}\right). \tag{34}$$

Therefore, the explicit forms of the effective potentials belonging to the SABH spacetime for spin-$\frac{3}{2}$ fermions are written as

$$V_{1,2} = F(r)\frac{\bar{\lambda}(1 - \bar{\lambda}^2)}{r^2(F - \bar{\lambda}^2)^2}\left[\pm\left(\frac{rF' - 2F}{2\sqrt{F}}\right)(F - \bar{\lambda}^2) \mp r\sqrt{F}F' + \bar{\lambda}(1 - \bar{\lambda}^2)\right]. \qquad (35)$$

In Equation (35), a prime symbol indicates a derivative with respect to $r$. In Figure 3, the behaviors of the effective potentials (35) of spin-$\frac{3}{2}$ fields propagating in the SABH geometry are illustrated for various $\zeta$ parameters.

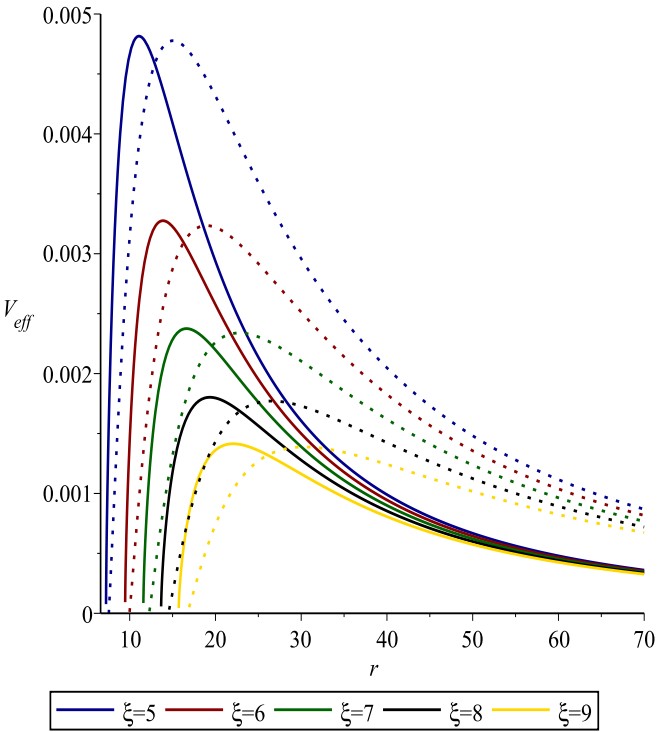

**Figure 3.** $V_{eff}$ versus $r$ graph for the spin-$\frac{3}{2}$ fermions. The plots are governed by Equation (35). The physical parameters are chosen as $M = j = 1$.

## 5. GFs of SABH via Fermion Emission

GFs in four-dimensional curved spacetime describe the absorption and scattering of fields by BHs. They play a crucial role in understanding the properties of BHs and their interactions with the surrounding environment. The concept of GFs was first introduced in the context of general relativity and quantum field theory, where they were used to study the emission and absorption of particles by BHs. The GFs depend on the geometry of the BH, the nature of the field, and the energy and angular momentum of the particles. In four-dimensional curved spacetime, the GFs can be calculated using a combination of analytical and numerical methods, including the WKB approximation, the partial wave analysis, and the Monte Carlo simulations. The calculation of GFs requires a proper treatment of the interaction between the field and the BH, taking into account the effects of the curvature of the spacetime and the presence of the event horizon. Several studies [77–79] have shown that the GFs can provide important information about the thermodynamics, stability, and quantum properties of BHs. They can also be used to study the evolution and dynamics of BHs, as well as their interactions with other objects in the universe. Overall, the literature on GFs in four-dimensional curved spacetime provides a rich and diverse field of study, with numerous insights and contributions to the understanding of BHs and their role in the universe.

In short, GFs for BHs are a measure of how much the spectrum of radiation emitted by a BH deviates from that of a perfect black body. The general semi-analytic bounds for the GFs are given by [13] (and see also Chandrasekhar's famous monograph [66] for the details).

$$\sigma(\omega) \geq \sec h^2 \left[ \int_{-\infty}^{+\infty} \wp \, dr_* \right],$$
(36)

where

$$\wp = \frac{\sqrt{(h'^2) + (\omega^2 - V_{eff} - h^2)^2}}{2h},$$
(37)

in which $h(x) > 0$ seen in the integrand of Equation (36) is some positive but otherwise arbitrary once-differentiable function [80]. We have two conditions for the certain positive function $h$ : (1) $h(r_*) > 0$ and (2) $h(-\infty) = h(+\infty) = \omega$. After applying the conditions to the effective potentials, one may observe a direct proportionality between the GFs and the effective potential, where the metric function plays a significant action in this process. After utilizing the aforementioned conditions and with the usage of the tortoise coordinate, Equation (36) becomes

$$\sigma(\omega) \geq \sec h^2 \left[ \int_{r_h}^{+\infty} \frac{V_{eff}}{2\omega F(r)} dr \right].$$
(38)

Since we have two types of fermions, we shall make the GFs computations in two cases.

### 5.1. Spin-$\frac{1}{2}$ Fermions

Spin-$\frac{1}{2}$ fermions, also known as Dirac fermions, play a crucial role in the description of quantum field theory and quantum mechanics. In quantum field theory, spin-$\frac{1}{2}$ fermions are used to describe the behavior of fundamental particles such as electrons, neutrinos, and quarks. One important aspect of the behavior of these particles is the concept of GFs, which describes the probability of the particles being scattered or absorbed by a gravitational or electromagnetic field. The study of GFs for spin-$\frac{1}{2}$ fermions is important because it provides insight into the interaction between these particles and their environment. For example, the GFs of electrons in a BH can be used to describe how the BH affects the electrons and their energy states. This understanding can then be used to make predictions about the behavior of these particles in different environments and help us better understand the behavior of the universe as a whole. One of the references that supports the importance of spin-$\frac{1}{2}$ fermion GFs is the paper "QNMs and GFs of the novel four-dimensional Gauss–Bonnet BHs in asymptotically de Sitter spacetime: scalar, electromagnetic and Dirac perturbations" by S. Devi et al (2020) [81]. This paper provides a comprehensive study of GFs for various particle types and their interactions with BHs. It provides a detailed mathematical analysis of the scattering and absorption of particles in a BH environment, including spin-$\frac{1}{2}$ fermions. Another reference that supports the importance of spin-$\frac{1}{2}$ fermion GFs is the paper "BHs in the quantum universe" by S. B. Giddings (2019) [82]. This paper provides an overview of the quantum mechanical properties of BHs and the interactions between BHs and particles. It discusses the importance of studying the GFs of spin-$\frac{1}{2}$ fermions and how this understanding can help us better understand the behavior of BHs and other astronomical objects. In conclusion, spin-$\frac{1}{2}$ fermion GFs play a critical role in the understanding of particle behavior in gravitational and electromagnetic fields. The study of these factors provides insight into the interaction between particles and their environment and can help us make predictions about the behavior of the universe as a whole.

Substituting the effective potential (22) derived from Dirac equations into Equation (38), we obtain

$$\sigma(\omega) \geq \sec h^2 \left[ \frac{1}{2\omega} \int_{r_h}^{+\infty} \frac{|k|}{r^2} \left( |k| \pm \left( \frac{r}{2\sqrt{F}} \frac{dF(r)}{dr} - \sqrt{F(r)} \right) \right) dr \right]. \tag{39}$$

After evaluating integral (39), the GFs of spin-$\frac{1}{2}$ fermions are found out to be

$$\sigma_l(\omega) \geq \sec h^2 \left[ \frac{|k|}{2\omega} \left( \frac{(k-1)}{r_h} + \frac{M(1+\xi)}{r_h^2} + \frac{2M^2}{3r_h^3}(1 + \xi^2 - \frac{9}{2}\xi) + \frac{M^3}{2r_h^4}(\xi^3 - 5\xi^2 + 3\xi + 1) \right) \right]. \tag{40}$$

GFs with varying $\xi$ parameters of perturbed SABH for the spin-$\frac{1}{2}$ fermions are depicted in Figure 4. The increase in $\xi$ value also increases the GF value for both spin-$\pm\frac{1}{2}$ fermions.

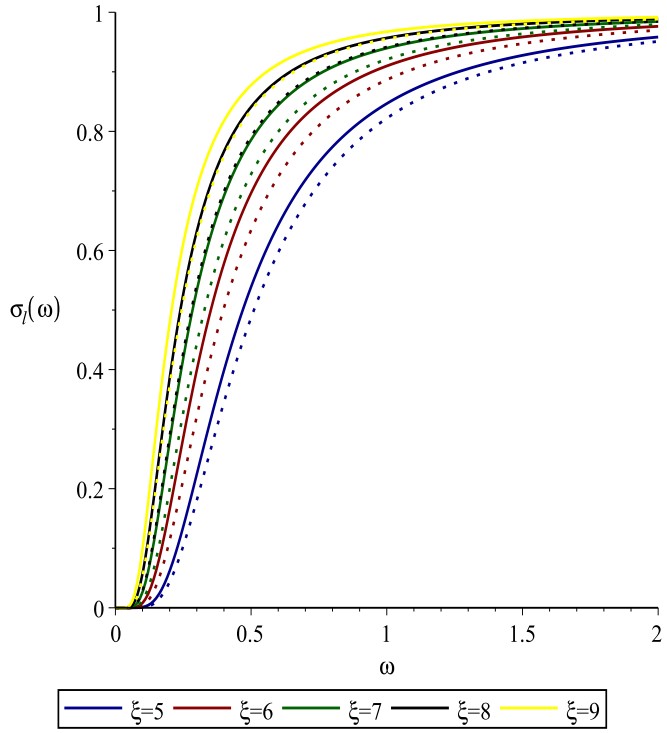

**Figure 4.** $\sigma_l(\omega)$ versus $\omega$ graph for the spin-$\frac{1}{2}$ fermions. While the solid lines stand for spin-$\frac{+1}{2}$ fermions, the dotted ones represent spin-$\frac{-1}{2}$ fermions. The physical parameters are chosen as $M = 1$ and $k = 2.5$.

## 5.2. Spin-$\frac{3}{2}$ Fermions

Spin-$\frac{3}{2}$ fermions, also known as Rarita–Schwinger particles, play a crucial role in high-energy physics and theoretical physics. The importance of spin-$\frac{3}{2}$ fermions GFs lies in the fact that they can be used to study the properties of exotic and massive particles, such as gravitons and gravitinos, in various physical situations. Another important application of spin-$\frac{3}{2}$ fermions GFs is in the study of the early universe. During the early universe, the production of massive particles such as gravitons and gravitinos played a crucial role in shaping the evolution of the universe. The GFs of spin-$\frac{3}{2}$ fermions have been studied in the context of the early universe and have been used to determine the impact of these massive particles on the evolution of the universe. Another important application of spin-$\frac{3}{2}$ fermions GFs is in the study of BH physics, as we do in this current study. In conclusion, the study of spin-$\frac{3}{2}$ fermions GFs is important for a better understanding of the properties of

exotic and massive particles in various physical situations. An interested reader is referred to some relevant references [76,83–85] in this area.

We now consider the GFs of SABH via the spin-$\frac{3}{2}$ fermions. To this end, we use the effective potential (35) in Equation (38) and obtain

$$\sigma(\omega) \geq \sec h^2 \left[ \frac{1}{2\omega} \int_{r_h}^{+\infty} \bar{\lambda}(1-\bar{\lambda}^2) \left( \pm \frac{rF' - 2F}{2r^2\sqrt{F}(F-\bar{\lambda}^2)} + \frac{\mp r\sqrt{F}F' + \bar{\lambda}(1-\bar{\lambda}^2)}{r^2(F-\bar{\lambda}^2)^2} \right) dr \right]. \tag{41}$$

To overcome the difficulties while evaluating the above complicated integral (41), the asymptotic series expansion method is applied. After performing straightforward calculations, one can obtain the following GFs for the spin-$\frac{3}{2}$ fermions emitted from the SABH:

$$\sigma(\omega) \geq \sec h^2 \left[ \frac{\bar{\lambda}(1-\bar{\lambda}^2)}{2\omega} \left( \frac{\bar{\lambda}(1-\bar{\lambda}^2)-1}{r_h(1-\bar{\lambda}^2)^2} + \frac{(1+\xi)}{2r_h^2(1-\bar{\lambda}^2)^2}(1-2\bar{\lambda}^2 + \frac{4\bar{\lambda}}{1-\bar{\lambda}^2}) + \frac{1}{3r_h^3(1-\lambda^2)} \times \right. \right.$$
$$\left. \left. \left( \frac{3}{2}(1+\xi)^2 - 12\xi - \frac{4(1+\xi)^2 + 16\xi}{\bar{\lambda}^2-1} - \frac{2(1+\xi)^2(\bar{\lambda}^2-3)}{(\bar{\lambda}^2-1)^2} + \frac{4\bar{\lambda}}{(1-\bar{\lambda}^2)^2(4\lambda^2\xi+3\xi^2+2\xi+3)} \right) \right) \right]. \tag{42}$$

As can be seen from Figure 5, GFs increase with the $\xi$ parameter and vice versa. Moreover, it was observed that GFs of spin-$\frac{-3}{2}$ fermions are higher than the spin-$\frac{+3}{2}$ fermions. Therefore, one can conclude that thermal emission of Rarita–Schwinger fermions from the SABH separates the particles of different spin into separate beams. So, SABH spacetime acts as a device similar to the famous experiment that is about how electrons are measured in a Stern–Gerlach magnetic field device, which splits up and down spin-$\frac{-3}{2}$ beams [86].

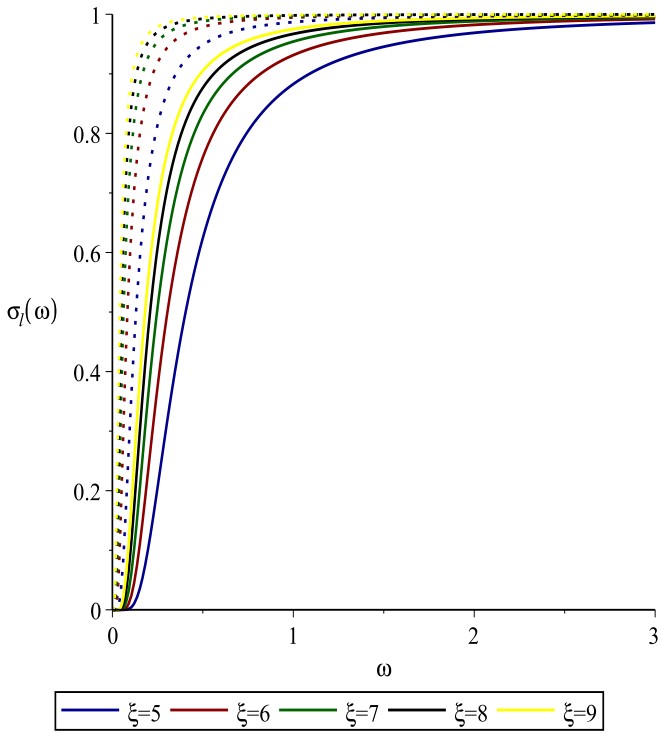

**Figure 5.** $\sigma_l(\omega)$ versus $\omega$ graph for the spin-$\frac{3}{2}$ fermions. While the solid lines stand for spin-$\frac{+3}{2}$ fermions, the dotted ones represent spin-$\frac{-3}{2}$ fermions; and also $\bar{\lambda} = 2$.

## 6. Conclusions

In this paper, we obtained analytical GFs for the covariant massless Dirac and Rarita-Schwinger equations in the SABH spacetime. The angular part of the solutions is given

in terms of the spherical harmonic functions, while the radial equations are reduced to one-dimensional Schrödinger-like wave equations. The analysis of the radial wave equations led to some interesting physics. We studied the thermal radiation (HR) spectrum for massless fermions in the vicinity to the exterior event horizon. Namely, we obtained the quasi-spectrum of greybody spectrum for massless fermions having spin-$\frac{1}{2}$ and spin-$\frac{3}{2}$ propagating in the SABH spacetime. We showed how the GFs values of this spectrum are found. This became possible by employing semi-analytic bounds for the GFs. We examined the behaviors of the obtained effective potentials and showed that $\zeta$ parameter modifies both the effective potentials and therefore the GFs. We showed that with the increased value of the $\zeta$ parameter, the GFs increase as well. This means that higher acoustic values of SABH will result in a higher probability of detecting HR. Moreover, it was shown that the thermal emission of Rarita–Schwinger fermions from a SABH result in the separation of fermions with different spin into distinct thermal radiations. Therefore, we presented some analytical results (Equations (40) and (42)) that might be compared with data to be detected in future. Finally, it is important to note that previous observations have confirmed the existence of HR in an analog BH [87]. Additionally, the wave phenomena examined in this research are caused by the interaction between quantum fields, specifically the fermion field, and the effective geometry of acoustic BHs in the Schwarzschild spacetime. Thus, they are intriguing semi-classical phenomena that can provide a deeper understanding of BH physics and therefore should be further studied in the near future.

**Author Contributions:** Conceptualization, İ.S. and S.K.; methodology, S.K.; software, S.K.; validation, İ.S. and S.K.; formal analysis, İ.S.; investigation, İ.S.; resources, İ.S.; data curation, S.K.; writing—original draft preparation, S.K.; writing—review and editing, İ.S.; visualization, S.K.; supervision, İ.S.; project administration, İ.S.; All authors have read and agreed to the published version of the manuscript.

**Funding:** This research received no external funding.

**Data Availability Statement:** Data sharing not applicable.

**Acknowledgments:** We would like to thank B. Pourhassan for useful conversation and suggestions related to the topic of the present paper. We also acknowledge the contributions of TÜBİTAK and SCOAP3. Authors would also like to acknowledge the contribution of the COST Action CA18108 and its researchers.

**Conflicts of Interest:** The authors declare no conflict of interest.

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
