# Peer review of "Fermionic Greybody Factors in Schwarzschild Acoustic Black Holes"

_universe, doi:10.3390/universe9020108_

Round 1

Reviewer 1 Report

This is a good and interesting paper, concerning the fermionic greybody factors for the acoustic Schwartzschild black holes. This paper is generally very well written, very well organized and it seems to me to be mathematically correct, thus this is a paper, which eventually deserves to be published. 

However, before it is ready for publication, I have a few questions and comments, for the authors of this paper: 

1. There is much confusion around equation (4), I simply don't understand what it means? In the round bracket, is it $\sin four dimensionali$? Come on, you cannot just mix Latex and ordinary words, that makes no sense! This point must be clarified, by the authors! 

2. Why at all introduce this $c_s$ squared, when it cancels out anyway, in equation (4) also? 

3. Also at page 4, in an equation inserted in the text, but at page 5, in equation (9). I have some doubts, that this equation is mathematically correct, because it means that the entropy becomes imaginary for small values of the parameter, and generally complex, can this really be true? 

4. What is positive, is that the authors of this paper, have provided an excellent Conclusion Section. As a frequent referee, I often see poorly written Conclusion Sections, but this is excellent! 

Further revision is necessary. Major revision. 

Author Response

Please see the attached pdf file. 

Reviewer 2 Report

Authors obtained analytical greybody factors for the covariant massless Dirac and Rarita–Schwinger equations in the Schwarzschild acoustic black holes spacetime.

In general the text of the manuscript looks like suitable for Universe journal.

Nevertheless, few questions arise.

(i) What are the names of the massless fermions having spin-1/2 and spin 3/2 propagating in the SABH spacetime? Maybe massless phonons?

(ii) Of course, groups of citations as [18–27] or [28–48] are possible. But together with the list of references without titles of the articles, as it is here, the information is going to zero. Important information is as follows. Which papers are about spin 3/2? Which articles are about spin ½? Which are about ordinary BH? Which are about SABH?

(iii) Rho meson, mentioned in text as Rarita–Schwinger equationis used in the study of high-energy physics and in the description of some subatomic particles, such as the Rho meson”, is the particle having spin 1. Please demonstrate in this line the relation of Rarita–Schwinger equation for the spin 1/2 and, of course, for spin 3/2 particles. Maybe the article “On the choice of relativistic wave equation for the particle having spin s = 3/2, J. Phys. Commun. 6 (2022) 075008” can be useful? Moreover, Rarita–Schwinger equation is not only one for the spin 3/2 particles description.

(iv) There are many articles about difficulties of Rarita–Schwinger equation, especially in the case of interaction…Why you are so simple thinking in this point?

(v) Laboratory experiment in this case is possible. Please find few words about it. I know that it is difficult for ordinary theoretical physics.

I am ready to accept this manuscript for publication, but today I can suggest the major revision.

Author Response

Please see the attached pdf file. 

Reviewer 3 Report

In this manuscript, the authors analyze the greybody factors (GF) of fermionic (1/2 and 3/2 spin) particle emission in Schwarzschild Acoustic black holes (SABH). In order to do so, they first compute the effective potential for different values of the acoustic tuning parameter, and then lower bounds of the GFs out of it.

The articles is poorly written and contains serious flaws. The main weaknesses can be classified in a confusing presentation of the physical scenario under consideration, and a lack of proper justification and connection of the results presented. In this circumstances, it is impossible to judge the ultimate meaning, not to say the validity, of the results. What it is clear to me, is that they cannot be supported with the material presented by the authors. If there were one or two relevant but specific problems, a major revision could address them. But in the present case the problems are almost all along the presentation, as I will summarize below. Therefore, I cannot recommend the paper for publication.

A (non-exhaustive) list of specific flaws and weaknesses. The main ones, which made me to reject the paper, are marked with (*):

- The previous results of refs. [14, 16] mentioned in the introduction (p2. lines 38-42) are not for general emission of black holes, but only specific to extremal ones. Not mentioning this is very confusing. There is also a blatant typo when writing "Hawking Radiation" (line 39).

- Page 3, lines 53-4. A white body is not an object that "would emit EM radiation at all frequencies, and at the same intensity for all frequencies". This would mean infinite energy emission. It is rather a body that reflects all incident radiation, as opposed to the black body that absorbs it all. This is a major conceptual mistake.

* Section II. When introducing the SABH geometry, the authors first write the action in eq. (1) and the wave function derived out of it (2), which then they do not use any more. What does \varphi mean within the article? What is it used for in the upcoming calculations?

* Right after this, the authors suddenly consider "the background spacetime metric as the Schwarzschild BH metric", eq. (3). What is the physical scenario that we are considering? Acoustic black holes "formed in certain type of fluids" (p. 1, line 19), or real black holes? If it is the former, which is the one I assumed, then the background metric is of course just Minkowski, not Schwarzschild. The very nature of the problem is unclear.

* Along the same line, the authors keep going introducing the acoustic metric (4) (where there is also a blatant typo), without any clear justification and without physical explanation of what the acoustic parameter \xi is. Moreover, in this metric they find an "optical [...] event horizon" (p. 4, lines 95-6). Are we then really in a true spacetime black hole? Because an acoustic black hole has nothing to do with light whatsoever. It is an effective metric for phonons. Again, the problem addressed is not clearly exposed.

* As an important remark with respect to the physical motivation: If the authors are considering a truly acoustic metric in some fluid, they should justify why is it interesting to consider fermionic fields in such scenario. Is there any perturbation of a fluid that behaves as a fermionic field? Or is it just a formal exercise to consider such fields in an acoustic metric?

- In FIG. 1, the authors do not mention the value of M that they consider.

- There is an important typo in eq. (11). It not should be equated to 0.

* The derivation of the effective potential in Section III is extremely confusing. For example, in eq. (15) the radial components G and H have a superscript (+/-) which disappears from there on. Also, it is not clear at all how the final effective potentials in (22) are derived out of the previous equations. To which components of the Dirac field does each potential correspond?

* In FIG 2. the authors plot the effective potentials for different values of \xi. However, the potentials in (22) also depend on M and most importantly on k, both of which are left unspecified. It is true that, for the potentials in (22), those values can be absorbed as normalizations of the potential and the radial coordinate. This is however not the case for the potentials in (35) and their dependence on j through \bar\lambda. This means that, if the results of the authors are correct, the shape of the potential differs for each value of the angular momentum of the mode considered. This is however not plotted in (35), where the value of j chosen is unknown.

* When computing the GF in Section V, the function h appearing in (36) remains a complete mystery. What is its meaning, where does it come from? How do the authors finally fix it to compute graphs later on?

* In FIGS. 4 and 5, values of the GF are plotted only depending on \omega and \xi. However, dependence on \omega does not show up in (40) or (42), which is odd. On the other hand, those expressions do depend, in strictly non-trivial ways, on k, \bar\lambda, M or r_h; and again, we do not know what values of these parameters do the authors consider. Moreover, in the expressions of the GF in (36), and therefore from there on, \sec h^2 appears as a global not-integrated factor. According to what is written below eq. (37), h is a function of r. Hence, the GF would be functions of the radius, which does not make any sense.

- The authors only compute lower bounds for the GF, but the graphs in FIGS. 4 and 5 assume that the very value of the GF is plotted.

* The confusion about what is the physical scenario under consideration remains equally present and unsolved in the conclusions in Section VI.

Author Response

Please see the attached pdf file. 

Round 2

Reviewer 1 Report

The authors of this paper, have answered all my questions and comments, which I raised in my first report, in a very satisfying way and they have improved their paper in a couple of different ways, so I can now clearly recommend this paper for publication in the Universe Journal of physics. 

Author Response

Dear Referee. Thank you very much. We are highly appreciated your suggestions and comments. Best Regards.

The Authors. 

Reviewer 2 Report

(i) Line 188. In any case spin 3/2 can not be  intermediate between spin 1/2 and spin 1. This idea should be explained in some details .

(ii) The comments of referee, who suggested rejection, are decisive.

I can suggest only formal acception.

Author Response

We realized that we could not clearly explain what we wanted to say. So the Referee is deginitely right obout his question "In any case spin 3/2 can not be  intermediate between spin 1/2 and spin 1. This idea should be explained in some details". What we wanted to say that spin-3/2 would be obtained like spin-1 and spin-1/2 from quarks' compositions. That's why we wrote it that way. But now we have corrected those phrases in the revised article so as not to leave any question marks in mind.
Our detailed answer to the Referee is as follows: 
As is well-known, quark is a type of elementary particle and a fundamental constituent of matter. All quarks are spin-half fermions  (s=1/2), have a fractional charge  (1/3  or  2/3e), and have baryon number  B=1/3 . Each quark has an antiquark with the same mass but opposite charge and baryon number. 
Quarks bind together in groups of two or three to form hadrons. Baryons are formed from three quarks. For example, the delta plus ( Δ+) baryon is formed from the same three quarks as the proton, but the total spin of the particle is 3/2 rather than 1/2. Similarly, the mass of  Δ+  with spin 3/2 is 1.3 times the mass of the proton, and the delta zero ( Δ0 ) baryon with a spin 3/2 is 1.3 times the neutron mass. Evidently, the energy associated with the spin (or angular momentum) of the particle contributes to its mass energy. 
The spin of the  (π+)   meson is 0. The same quark-antiquark combination gives the rho (ρ) meson with spin 1. This meson has a mass approximately 5.5 times that of the  π+   meson.
See for example:

https://www.icr.org/article/subatomic-particles-part-3-mesons

https://phys.libretexts.org/Bookshelves/University_Physics/Book%3A_University_Physics_(OpenStax)/University_Physics_III_-_Optics_and_Modern_Physics_(OpenStax)/11%3A_Particle_Physics_and_Cosmology/11.04%3A_Quarks

Reviewer 3 Report

-------------

Author Response

Dear Editor & Referee. Thank you very much. We are highly appreciated your suggestions and comments. Best Regards.

The Authors.